# Use of Nitric Oxide and Hydrogen Peroxide for Better Yield of Wheat (*Triticum aestivum* L.) under Water Deficit Conditions: Growth, Osmoregulation, and Antioxidative Defense Mechanism

**DOI:** 10.3390/plants9020285

**Published:** 2020-02-22

**Authors:** Noman Habib, Qasim Ali, Shafaqat Ali, Muhammad Tariq Javed, Muhammad Zulqurnain Haider, Rashida Perveen, Muhammad Rizwan Shahid, Muhammad Rizwan, Mohamed M. Abdel-Daim, Amr Elkelish, May Bin-Jumah

**Affiliations:** 1Department of Botany, Government College University, Faisalabad 38000, Pakistan; 2Department of Environmental Sciences and Engineering, Government College University, Faisalabad 38000, Pakistan; 3Department of Biological Sciences and Technology, China Medical University, Taichung 40402, Taiwan; 4Department of Physics, University of Agriculture, Faisalabad 38040, Pakistan; 5Institute of Soil and Environmental Sciences, University of Agriculture, Faisalabad 38040, Pakistan; hellorizwan@outlook.com; 6Department of Zoology, College of Science, King Saud University, P.O. Box 2455, Riyadh 11451, Saudi Arabia; abdeldaim.m@vet.suez.edu.eg; 7Pharmacology Department, Faculty of Veterinary Medicine, Suez Canal University, Ismailia 41522, Egypt; 8Botany Department, Faculty of Science, Suez Canal University Ismailia, Ismailia 41522, Egypt; amr.elkelish@science.suez.edu.eg; 9Department of Biology, College of Science, Princess Nourah bint Abdulrahman University, Riyadh 11474, Saudi Arabia; may_binjumah@outlook.com

**Keywords:** drought stress, antioxidative potential, biomass, grain yield, hydrogen peroxide, nitric oxide, seed priming, wheat, proline, glycinebetaine

## Abstract

The present experiment was carried out to study the influences of exogenously-applied nitric oxide (NO) donor sodium nitroprusside (SNP) and hydrogen peroxide (H_2_O_2_) as seed primers on growth and yield in relation with different physio-biochemical parameters, antioxidant activities, and osmolyte accumulation in wheat plants grown under control (100% field capacity) and water stress (60% field capacity) conditions. During soaking, the seeds were covered and kept in completely dark. Drought stress markedly reduced the plant growth, grain yield, leaf photosynthetic pigments, total phenolic content (TPC), total soluble proteins (TSP), leaf water potential (Ψ_w_), leaf turgor potential (Ψ_p_), osmotic potential (Ψ_s_), and leaf relative water content (LRWC), while it increased the activities of enzymatic antioxidants and the accumulation of leaf ascorbic acid (AsA), proline (Pro), glycine betaine (GB), malondialdehyde (MDA), and H_2_O_2_. However, seed priming with SNP and H_2_O_2_ alone and in combination mitigated the deleterious effects of water stress on growth and yield by improving the Ψ_w_, Ψ_s_, Ψ_p_, photosynthetic pigments, osmolytes accumulation (GB and Pro), TSP, and the antioxidative defense mechanism. Furthermore, the application of NO and H_2_O_2_ as seed primers also reduced the accumulation of H_2_O_2_ and MDA contents. The effectiveness was treatment-specific and the combined application was also found to be effective. The results revealed that exogenous application of NO and H_2_O_2_ was effective in increasing the tolerance of wheat plants under drought stress in terms of growth and grain yield by regulating plant–water relations, the antioxidative defense mechanism, and accumulation of osmolytes, and by reducing the membrane lipid peroxidation.

## 1. Introduction

Wheat is one of the major cereals in the world, being a main source of calories and protein in most parts of the world. However, its production is severely threatened by different abiotic factors, including drought. Shortage of fresh water affects various aspects of plants, including morphological factors, as well as several physiological and biochemical mechanisms, resulting in reduced final production [1,2,3]. Among different physiological mechanisms, water-stress-induced excessive production of reactive oxygen species (ROS), such as singlet oxygen (^1^ O_2_), hydrogen peroxide (H_2_O_2_), superoxide anions (O_2_^−^), and hydroxyl radical (OH^−^), are the important ones that damage the lipids, proteins, photosynthetic pigments, and nucleic acids [4]. Under severe stress, these damages eventually lead to death of cells, and finally of the plant [5]. At cellular levels, chloroplasts, vacuoles, microbodies, and mitochondria are the organelles or sites for production of ROS [6].

Plants have a well-developed antioxidative defense system (enzymatic and non-enzymatic) to detoxify and neutralize the excessively produced ROS [7,8,9]. Enzymatic antioxidants help in maintaining the defense in response to oxidative damage. Although superoxide dismutase (SOD) is the key constituent of the antioxidative defense mechanism, peroxidase (POD) and catalase (CAT) helps in neutralizing some potential oxidants, thus increasing the plant’s resistance under induced osmotic stress [6]. If the amount of antioxidants is reduced in response to stressful conditions, then the oxidative damages caused to the photosynthetic machinery result in ultimate cell death [10]. Among non-enzymatic components, carotenoids, ascorbic acid (AsA), and phenolics are considered as the most important ones among others due to their antioxidative and biological characteristics [11,12]. The contribution of ascorbic acid in the cellular antioxidative defense mechanism [7] has been described in the Halliwell–Asada pathway in detail. Studies revealed the enhancement in osmotic stress tolerance as a result of higher antioxidant activities in several plants, such as maize [13], radish [14], and wheat [15].

In order to overcome the damaging impacts of drought stress, plants have also developed the mechanism of osmotic adjustment through increased synthesis of osmolytes (glycine betaine (GB) and proline (Pro)), secondary metabolites, and carbohydrates [16,17]. Osmotic homeostasis plays an important role in maintaining plant growth and cell turgor by reducing osmotic potential, resulting in better growth [18]. Gou et al. [19] found that in maize plants under drought stress, accumulation of choline and GB was found to be effective in plant water states as a result of better growth. In parallel, drought-induced higher accumulation of osmoprotectants, such as total soluble proteins and proline, was effective for osmoregulation in cotton plants [9]. However, the stress tolerance mechanism in plants is species- and crop-cultivar-specific. Some high-yielding crop genotypes are sensitive to different abiotic stresses. Different methods or techniques are being used to improve the crop stress tolerance for better production [1,20,21]. The response of drought-treated *Brassica napus* plants was found to be positively associated with osmotic regulation [22].

Pre-sowing seed treatment with different inorganic and organic chemicals is considered an important one way to increase the plant tolerance to stressful conditions [7,23]. Of the different seed priming agents, H_2_O_2_ is one such compound. Although the H_2_O_2_ is well-known due to its damaging effects to the cellular membranes, it has been found that seed priming with H_2_O_2_ at lower levels can improve the stress tolerance of plants under stressful conditions [24]. It has been found that in many plant species, seed priming with H_2_O_2_ played a significant protective role in the induction of tolerance to oxidative stress caused by salinity, temperature, osmotic stress, as well as metal toxicity [23,25,26,27]. Furthermore, seed priming with H_2_O_2_ was found to be effective in improving the activities of enzymes of the antioxidative defense mechanism [28], as well as improving the stomatal regulation, biosynthesis of photosynthetic pigments, photosynthetic efficiency, biomass production, and formation of adventitious roots [29,30]. It has been reported that H_2_O_2_ priming contributes significantly in regulating the plant growth and physiology, revealing the improved resistance of plants to abiotic stresses [31]. Liu et al. [32] suggested that priming with H_2_O_2_ is effective in improving the activities of antioxidant enzymes in the leaves of cucumber plants under osmotic stress by reducing the malondialdehyde (MDA) accumulation and protecting the ultrastructure of mitochondria and chloroplasts. As a gaseous signaling molecule, H_2_O_2_ also regulates cellular water relations by osmotic adjustment through controlling the biosynthesis of cellular osmolytes, such as proline, glycinebetain, and soluble proteins. [33]. It has been reported that H_2_O_2_ stimulates the biosyntheses of proline in water-stressed seedlings of maize by controlling the activities of ornithine aminotransferase (OAT) and arginase, glutamate dehydrogenase (GDH), and ∆1-pyrroline-5-carboxylate synthetase (P5CS), with an alternate decrease in the activity of proline dehydrogenase [34].

Another priming agent, sodium nitroprusside (SNP, donor of nitric oxide, a gaseous signaling molecule), regulates different physio-biochemical and developmental processes of plants, such as root and shoot growth, along with their patterns [35,36], plant–water relations [37], floral regulation [38], seed germination [39,40], photosynthesis [28], stomatal conductance [41], fruit ripening, and senescence [38]. Being an antioxidant, it can also reduce the oxidative damage by detoxifying the over-produced ROS [42]. Nitric oxide has the potential to decrease the levels of H_2_O_2_ and lipid peroxidation under water stress [43]. Exogenously-applied nitric oxide (NO) also improves the plant biomass production and leaf chlorophyll biosynthesis by improving the activities of antioxidants to reduce the damages caused by oxidative stress [44,45]. Nitric oxide signaling also plays an important role in plant disease resistance [46] in response to several abiotic stresses [5,7,8,47]. Nitric oxide also plays a significant role in plant biomass production and yield by imparting its significant role in cellular water relations through controlling the cellular osmotic adjustment, in turn by controlling the biosynthesis of osmolytes [33]. In *Cakile maritima* seedlings grown under water deficit conditions after pretreatment with SNP, a donor of NO significantly enhanced the proline accumulation, and this increase in proline accumulation was due to increased activity of P5CS, a proline synthesizing enzyme [48]. This effect of SNP on proline biosynthesis was due to its signaling properties.

It has been reported that in combination, NO and H_2_O_2_ are both important as stress signaling molecules, mediating plant responses to different environmental constraints, including water stress, extremes of temperature and ultraviolet (UV) radiations, as well as biotic stresses [49,50]. In combination, they regulate the abscisic acid (ABA) signaling pathway for stomatal closure [51] and for the activation of antioxidant enzymes [50,52,53] in response to different abiotic stresses. Lin et al. [54] found that H_2_O_2_ promotes the production of NO by increasing the activities of nitrate reductase (NR) in the leaves of the *noe1* plant. Similarly, NO may also trigger the activities of enzymatic antioxidants to regulate the concentration of H_2_O_2_ [52]. Lin et al. [54] suggested that in marigold explants, both the NO and H_2_O_2_ were effective in protecting the ultrastructure of mesophyll cells and in increasing the leaf photosynthetic performance in response to osmotic stress in adventitious rooting. Likewise, the signaling interaction among NO and H_2_O_2_ enhanced the myo-inositol phosphate synthase activity to mitigate the adverse effects of osmotic stress [55]. Additionally, Luna et al. [50] proposed that endogenous H_2_O_2_ and NO may contribute in ABA-triggered drought resistance of Bermuda grass, improving the activities of enzymatic antioxidants. Though literature depicts that the interplay among NO and H_2_O_2_ reduces the deleterious impacts of the osmotic stress by activating antioxidative defense and physio-biochemical mechanisms, no or very few reports are available about the role of these signally molecules in yield increments in combination with different physio-biochemical attributes, alone or in combination. Therefore, the present study was planned with the objectives of finding out the extent to which the exogenously-applied NO and H_2_O_2_, separately as well as in combination, are helpful in reducing the water-stress-induced adverse effect on yield of wheat plants through the regulation of plant growth, water relations, and osmotic adjustment for better water relations, as well as the regulation of the antioxidative defense mechanism.

## 2. Materials and Methods

Experiments were done in a greenhouse in the Botanical Garden of Government College University, Faisalabad, Pakistan, to study the effects of seed priming with SNP and H_2_O_2_ separately and in combination (SNP + H_2_O_2_) on two different drought-tolerant wheat cultivars (Fsd-2008 and S-24) grown under drought stress, focusing on various morphological, physiological, biochemical, and yield parameters. Seeds of wheat cultivar (cv.) Fsd-2008 were obtained from Ayub Agricultural Research Institute (AARI) and those of cv. S-24 were obtained from the Department of Botany, Government College University, Faisalabad. Both of these cultivars are high yielding and are frequently used by breeders to develop new wheat genotypes. Cultivar S-24 is moderately drought-tolerant, but the cv. Fsd-2008 is highly drought-tolerant. Seeds of both wheat cultivars were soaked in different concentrations of SNP and H_2_O_2_ for 16 h, then tagged as T2 (SNP (0 and 0.1 mM)), T3 (H_2_O_2_ (0 and 1 mM)), or T4 (SNP + H_2_O_2_ (0.1 mM and 1 mM)), while the non-treated ones were tagged as T1. During soaking period, the seeds were continuously kept in darkness. After sowing, the seeds were air-dried until a constant weight was obtained. Ten seeds were sown per replicate in plastic pots (28 to 30 cm, filled with 10 kg sandy clay loam soil). The design of the experiment was a completely randomized design (CRD), with four replications of each treatment. Four plants in each pot were maintained by thinning after the eight days of germination. The up-rooted seedlings were crushed and mixed in soil in the same pot to maintain treatment uniformity. There were two sets of pots (i.e., one set was supplied with normal irrigation and another set was taken as water-stressed). After eight days of thinning of plants, water stress (60% field capacity) was maintained by withholding watering. To maintain the required field capacity, soil moisture content was monitored on a daily basis based on soil water tension measured using a tensiometer (Irrometer, Model, LT-12 inch, Riverside, CA). The 60% field capacity was maintained based on tensiometric value (i.e., 1 KPa = 1 mm of water). The requirement of N, P, and K fertilizers was fulfilled using urea, diammonium phosphate (DAP), and sulphate of potash (0.03:0.025:0.025 g/kg of soil), respectively. The average daily environmental conditions were recorded as follows: the average D/N length was 14/10 h, minimum and maximum D/N temperature was 18–9 ± 2.5 and 25–15 ± 3 °C, respectively, and average daytime relative humidity was 50%.

Data was recorded after six weeks of the application of water stress for various growth and physio-biochemical parameters. Data for different growth attributes such as root and shoot fresh weights was measured at the vegetative stage. After measuring fresh weights, shoots and roots were then kept at 70 °C for 48h for the estimation of dry weights. At the same growth stage, fresh leaves were collected and preserved at −80 °C for the estimation of different biochemical attributes. Two plants in each pot or replicate were kept growing under above mentioned water-stressed conditions until maturity for the estimation of different yield parameters.

### 2.1. Plant Water Relations

*Estimation of leaf water potential (Ψ_w_), leaf turgor potential (Ψ_p_), osmotic potential (Ψ_s_).* To determine leaf Ψw, a fully grown top second leaf at the vegetative stage was removed and a Scholander-type pressure chamber Model 615D, PMS Instruments, USA (between 6 a.m. to 8.30 a.m.) was used for the estimation of Ψw. The same leaves were then frozen at −20 °C for one week in a freezer to estimate the Ψs. The Ψs of the frozen leaves was estimated by using an osmometer (Wescor 5500). For the estimation of Ψp, the equation given below was used as reported by [56]:Ψp = Ψw − Ψs(1)

*Estimation of leaf relative water content**(LRWC).* LRWC was measured according to the method of [57]. Briefly, a fully mature 2nd leaf from the top was separated, the fresh weight was measured, and then soaked for 4 h in distilled H_2_O_2_. The soaked leaves were then removed from water, the excess water on the leaf surface was absorbed using filter paper, and the turgid weight was measured. Then, these leaves were dried using an oven at 65 °C for 48 h and their dry weights were measured. The formula given below was used for the estimation of leaf RWC:

Fresh weight of leaf—Dry weight of leaf
LRWC (%) = ---------------------------------------------------------------- × 100

Turgid weight of leaf—Dry weight of leaf

*Estimation of yield parameters*. Different yield parameters of wheat plants were estimated at maturity. After counting the number of tillers, spikes and grains were separated. Then, the grain yield per plant and 100 grain weight were estimated.

*Estimation of leaf photosynthetic pigments*. Leaf photosynthetic pigments such as chlorophyll *a* (Chl. *A*), Chl. *b*, total Chl., and Chl. *a/b* ratio were estimated using the method of [58], using acetone as an extractant. Briefly, fresh leaf material (0.1 g) was chopped and put into 10 mL of 80% acetone overnight at 4 °C. The absorbance of the extractant was read at 663, 645, and 480 nm by using a spectrophotometer (Hitachi U-2001, Tokyo, Japan). The formulas given below were used for the estimation of specific content of the photosynthetic pigments.
Chl. *a* = [12.7(∆A 663) − 2.69 (∆A 645)] × V/1000 × W
Chl. *b* = [22.9 (∆A 645) − 4.68 (∆A 663)] × V/1000 × W
Total Chl. = [20.2 (∆A 645) − 8.02 (∆A 663)] × V/1000 × W

### 2.2. Activities of Enzymatic Antioxidants

*Extraction of enzymes*. For the extraction of enzymes and total soluble protein (TSP), already preserved fresh leaf material at −80 °C was used. Briefly, the fresh leaf material (0.5 g) was ground well in pre-chilled phosphate buffer (10 mL) at pH 7.8. During the extraction, the pestle and mortar were kept in ice. The obtained extract was then centrifuged at 20,000× *g* for 20 min. The supernatant was then preserved at −80°C and used for the estimation of activities of enzymes such as CAT, POD, and SOD.

*Activity of SOD.* The method ascribed by Giannopolitis et al. [59] was used for the estimation of SOD activity. It works on the principle that SOD inhibits the photochemical reduction of nitroblue tetrazolium (NBT). The reaction solution (1 mL) comprised 50 mM phosphate buffer (pH 7.8), 13 mM methionine, 50 µM nitro blue tetrazolium (NBT), 1.3 µM riboflavin, and 50 µL enzyme extract. After mixing well, the cuvettes containing 1 mL triturate were irradiated for 15 min under fluorescent light (20 W) with photon flux density of 78 µmol m^−2^ s^−1^. The optical density of the reaction mixture was then read at 560 nm using a UV-visible spectrophotometer (Hitachi U-2100; Hitachi, Tokyo, Japan). During each estimation, a blank sample without the enzyme extract was used. One unit of SOD activity was defined as the amount of enzyme that caused 50% inhibition in photochemical reduction of NBT.

*Activities of POD and CAT.* The method described by Chance and Maehly [60] was followed for the estimation of activity of POD and CAT. The buffer extracted samples were used for this purpose. The reaction solution (3 mL) contained 59 mM H_2_O_2_, 0.1 mL sample extract, and 50 mM phosphate buffer (pH 7.8), which was used for the estimation of CAT activity. After initiation of reaction by the addition of extract, the change in optical density was read at 240 nm for 2 min at intervals of 20 s. The reaction mixture for POD activity was prepared by adding 40 mM H_2_O_2,_ 20 mM guaiacol, 50 mM phosphate buffer, and 0.1 mL enzyme extract. The optical density of the reaction mixture was read at 470 nm for 2 min at an interval of 20 s. The activity of the enzyme was expressed on a protein weight basis.

*Estimation of leaf ascorbic acid (AsA) content.* The method ascribed by Mukherjee and Choudhuri [61] was followed for the estimation of leaf AsA content. Fresh leaf samples (0.25 g) as preserved at −80 °C were ground in 6% trichloroacetic acid (TCA) (10 mL) solution. After centrifugation, 4 mL of the supernatant was mixed with 2% dinitrophenyl hydrazine solution (prepared in acidic medium (2 mL). Two drops of 10% solution of alcoholic thiourea (prepared in 70% ethanol) were added to the mixture. The triturate was then boiled for 20 min in a water bath. Then, 5 mL of 80% H_2_SO_4_ (*v/v*) were mixed with triturate after cooling. The samples during that process were kept at 0 °C. Samples were then read spectrophotometricaly at 530 nm. The quantity of AsA in the extracted leaf samples was worked out from a standard curve, prepared using a range of AsA standards.

*Estimation leaf total phenolics*. The method ascribed by Julkunen-Tiitto [62] was used for the estimation of total phenolic content in leaf samples. Briefly, leaf samples (0.5 g) were homogenized in 80% acetone. The supernatant obtained after centrifugation at 1000× *g* for 10 min was used. An aliquot (100 µL) was reacted with 2 mL of distilled water and 1 mL of Folin–Ciocalteau’s phenol reagent. The triturate was then mixed with 5.0 mL of 20 % Na_2_CO_3_ solution and the final volume of the mixture was measured to 10 mL with distilled H_2_O. After mixing well with a vortexer, the absorbance of the final solution was read at 750 nm using a UV-visible spectrophotometer (Hitachi U-2100).

*Free Proline Determination*. The method ascribed by Bates et al. [63] was used for the estimation of free proline in leaves. An amount of 0.5g of fresh leaf was homogenized in 3% solution of sulfo-salicylic acid (10 mL). Then, the 2.0 mL of the homogenate after filtration was mixed thoroughly with 2.0 mL of acid ninhydrin, which was prepared by adding 1.25 g ninhydrin in 20 mL of 6 M orthophosphoric acid and 30 mL glacial acetic acid, along with 2 mL of glacial acetic acid in a test tube. The mixture was then incubated for 60 min at 100 °C. The final material was then cooled immediately in an ice bath. Then, the triturate was mixed with 4.0 mL of toluene using a vortexer. The toluene layer containing chromophore was separated and kept at room temperature for a few minutes and the absorbance was read at 520 nm on a spectrophotometer using toluene as a blank. The concentration of proline was estimated using a standard curve prepared from a range of standards (0–50 mg/kg), such as pure proline, and the specific content was calculated on a fresh weight basis as follows:
µmole proline g^−1^ fresh weight = (µg proline mL^−1^ × mL of toluene/115.5)/(g of sample)

*Estimation of GB content*. The method ascribed by Grieve and Grattan [64] was used for estimation of GB content. Already dried (dried for 72 h in an electric oven at 65 °C) leaf material (1.0 g) was ground in distilled water (10 mL) and filtered. The filtrate (1 mL) was mixed thoroughly with 1 mL of 2N HCl. Then, 0.5 mL of triturate was taken in a glass tube and mixed with potassium iodide (KI) solution (0.2 mL). The final mixture was then shaken well and cooled in an ice bath for 90 min with continuous shaking. The resultant material was then mixed with 20 mL of 1–2 dichloromethane (cooled at −10 °C) and 2.0 mL of ice-cooled distilled water. The two layers formed in the mixture were mixed by passing a continuous stream of air for 1–2 min while tubes were still in an ice bath (4 °C). The optical density of the organic layer was measured at 365 nm after discarding the upper layer. A standard curve prepared from pure betain was used for the estimation of the betain amount in the samples.

*Estimation of TSP*. TSP in the extracted buffer was estimated following the method ascribed by Bradford [65]. An aliquot of 100 µL was mixed well with 2 mL of Bradford reagent. The absorbance of the mixture was read at 595 nm and the content of the TSP in samples was computed using a series of protein standards (200–1400 mg/kg), prepared from analytical grade bovine serum albumin.

*Estimation of H_2_O_2_ content.* The method ascribed by Velikova et al. [66] was followed for the estimation of H_2_O_2_ content. Briefly, fresh leaf samples were homogenized in TCA (6%). The material was centrifuged at 1000× *g* and then 0.1 mL of supernatant was mixed with 1 mL of KI. The absorbance of the resultant material was measured at 390 nm.

*Estimation of MDA content.* The method ascribed by Cakmak and Horst [67] was followed for the estimation of leaf MDA content. Fresh leaf samples (1.0 g) were homogenized in 0.1% TCA solution (3 mL). The homogenate was centrifuged for 15 min at 20,000× *g*. Then, 0.5 mL of the supernatant was mixed with 3 mL of 0.5% thiobarbituric acid (TBA) solution prepared in 20% TCA. The resultant mixture was then heated in a shaking water bath for 50 min at 95 °C. After centrifugation at 10,000× *g* for 10 min, the absorbance of the supernatant was read at 532 and 600 nm using the spectrophotometer. The MDA content was estimated as the difference in absorbance at 600 and 532 nm using the following formula:
MDA level (nmol) = Δ (A 532 nm-A 600 nm)/1.56 × 105

Absorption coefficient for calculating MDA is 156 mmol^−1^ cm^−1^.

### 2.3. Statistical Analysis

The experiment was laid out in a completely randomized design (CRD) with three replicates of each treatment, and the data for different attributes was analyzed using a software named CoSTAT Window’s version 6.3 (developed by Cohort software, Berkeley, California, USA) to study the significant differences among the treatments. Least significance difference (LSD, at 5% level) test was used to find out the significant differences among mean values of treatments. Principle component analysis (PCA) and correlation studies of the studied attributes were computed using the XLSTAT software version 4.15.

## 3. Results

### 3.1. Different Growth Attributes of Water-Stressed Wheat Plants Grown from Seeds Treated with Different Levels of SNP and H_2_O_2_

A significant reduction (*p* ≤ 0.001) was recorded in fresh and dry weights of shoots, as well as the lengths of shoots and roots of both studied wheat cultivars when grown under water stress. Seed priming with SNP and H_2_O_2_ separately, as well as their combined application (SNP + H_2_O_2_), significantly improved the fresh and dry weights, as well as the lengths of shoots and roots of wheat plants of both cultivars; slightly more improvement was found in plants of cv. Fsd-2008 under both watered-stressed and non-stressed conditions. Among the treatments, H_2_O_2_ seed priming was more effective as compared with other treatments in improving the root fresh and dry biomass and root lengths, while maximum improvement in shoot length was recorded in plants grown from seeds primed with combined application (SNP + H_2_O_2_) (Figure 1A,B,E,F; Table 1).

Shoot dry and fresh biomasses of both studied wheat cultivars also decreased significantly (*p* ≤ 0.001) when grown under water-stressed conditions. However, seed priming with SNP and H_2_O_2_ alone, as well as in combination (SNP + H_2_O_2_), significantly (*p* ≤ 0.001) improved the fresh and dry biomass of shoots of both wheat cultivars when grown under non-stressed and water-stressed conditions. A slightly greater increase in shoot fresh and dry weights was found in cv. S-24 as compared with cv. Fsd-2008, both under water-stressed and non-stressed conditions. Of different treatments, the combined application (SNP + H_2_O_2_) was found to be more effective as compared with the individual treatments of SNP and H_2_O_2_, respectively, for increasing the fresh and dry biomasses of shoot under both stressed and non-stressed conditions (Figure 1C,D; Table 1).

### 3.2. Photosynthetic Pigments and Activities of Antioxidant Enzymes of Water-Stressed Wheat Plants Grown from Seeds Primed with Different Levels of SNP and H_2_O_2_

Drought stress considerably decreased the Chl. *a*, *b*, and total Chl. contents in both cultivars of wheat. Seed priming with different treatments was helpful in improving the content of leaf photosynthetic pigments in both cultivars of wheat under both water-stressed and non-stressed conditions. Of the different treatments, H_2_O_2_ seed priming was more effective for improving Chl. *a* content, while for Chl. *b* content, SNP seed priming was more effective under both water-stressed and non-stressed conditions. However, regarding the leaf total Chl., the individual treatments of H_2_O_2_ and SNP were equally effective, followed by their combined treatment under water-stressed conditions (Figure 2A–C; Table 1).

A significant (*p* ≤ 0.001) improvement in the activities of POD and CAT in both wheat genotypes was recorded when under water deficit conditions, and this improvement was comparatively greater in cv. S-24 as compared with cv. Fsd-2008. Seed priming with SNP and H_2_O_2_ individually, as well as their combined application (SNP + H_2_O_2_), further enhanced the CAT and POD activities in both studied cultivars, and a slightly greater increase was found in cv. S-24 under drought-stressed conditions. For POD activity, SNP treatment only was more effective, followed by H_2_O_2_ under water-stressed conditions, while the combined treatment (SNP + H_2_O_2_) was more helpful in improving the CAT activity (Figure 2E,F; Table 1).

Similar to the activities of CAT and POD, the activity of SOD was also increased considerably in both studied wheat cultivars when exposed to water-stressed conditions, and slightly more improvement in leaf SOD activity was found in cv. Fsd-2008 (19%) in comparison with cv. S-24 (18%). Application of different treatments such as seed priming was helpful in further increasing the leaf SOD activity in both wheat cultivars under both non-stressed and water-stressed conditions. Among different treatments, SNP priming was most effective followed by H_2_O_2_ treatment under water stress in increasing the leaf SOD activity (Figure 2D; Table 1).

### 3.3. Leaf phenolics, AsA, GB, Proline, TSP, MDA, and H_2_O_2_ Contents of Water-Stressed Wheat Plants Grown from SNP and H_2_o_2_-Treated Seeds

A considerable increase in leaf AsA accumulation was recorded in both studied wheat cultivars due to water stress. Cultivar S-24 was superior in this regard as compared with cv. Fsd-2008 under water stress. Seed priming with SNP, H_2_O_2_, and SNP + H_2_O_2_ further improved the accumulation of AsA content in plants of both wheat cultivars. Comparatively greater accumulation was recorded in cv. S-24 as compared with cv. Fsd-2008. Among different treatments, H_2_O_2_ treatment alone was more effective, followed by SNP and SNP + H_2_O_2_, respectively. A similar improvement in leaf AsA content due to SNP and H_2_O_2_ was also recorded under non-stressed conditions (Figure 3A; Table 1).

Water stress considerably decreased the leaf phenolics accumulation in both wheat cultivars. This reduction induced by water stress was similar in wheat plants of both cultivars. Application of different treatments such as seed priming was helpful in increasing the leaf total phenolics in both wheat cultivars. Of the different treatments, SNP seed priming alone was more effective in comparison to other treatments under both water-stressed and non-stressed conditions (Figure 3B; Table 1).

A considerable improvement in leaf GB accumulation was recorded in water-stressed plants of both wheat cultivars, and greater accumulation was found in cv. Fsd-2008 (23.45%) as compared with cv. S-24 (22%). Seed pre-treatment with SNP and H_2_O_2_ alone as well as their combined application (SNP + H_2_O_2_) further increased the leaf GB content, but the increase was treatment- and cultivar-specific. Among different treatments, H_2_O_2_ seed priming was more helpful in improving the leaf GB content in both wheat cultivars followed by SNP treatment under both non-stressed and water-stressed conditions. However, regarding the combined treatment (SNP + H_2_O_2_), this improvement in GB was found only in plants of cv. Fsd-2008 under both non-stressed and water-stressed conditions (Figure 3C; Table 1).

Proline accumulation was also increased considerably in both wheat cultivars due to water stress. A further improvement in proline accumulation was recorded due to seed priming with SNP and H_2_O_2_ alone, as well as in combination (SNP + H_2_O_2_) in both wheat cultivars, under both non-stressed and stressed conditions. Among different treatments, SNP treatment alone was most effective, followed by H_2_O_2_ and combined treatment, respectively, under water stress (Figure 3D; Table 1).

Water stress also significantly (*p* ≤ 0.001) reduced the TSP accumulation in both wheat cultivars, and this reduction in TSP was comparatively less in cv. Fsd-2008. Exogenously-applied SNP and H_2_O_2_ as seed priming alone and in combination (SNP + H_2_O_2_) improved the TSP accumulation in both studied wheat cultivars under both stressed and non-stressed conditions. Among different treatments, SNP treatment alone was more effective as compared with other treatments. In comparison, greater accumulation in leaf TSP due to seed priming with SNP and H_2_O_2_ alone, as well as due to their combined treatment, was found in plants of cv. Fsd-2008 as compared with cv. S-24 under both non-stressed and water-stressed conditions (Figure 3E; Table 1).

Imposition of water-stress significantly enhanced the (*p* ≤ 0.001) leaf MDA accumulation in both cultivars of wheat. Seed priming with different treatments significantly (*p* ≤ 0.001) decreased the MDA accumulation in both cultivars of wheat under water stress. Among different treatments, the combined treatment (SNP + H_2_O_2_) was more effective, followed by solo treatments of H_2_O_2_ and SNP, respectively (Figure 3F; Table 1).

Leaf H_2_O_2_ levels were also increased considerably in wheat plants of both cultivars under water deficit conditions. Application of exogenous SNP and H_2_O_2_ alone and in combination (SNP + H_2_O_2_) was effective in reducing the leaf H_2_O_2_ levels in both studied wheat cultivars under both non-stressed and water-stressed conditions. Among all the treatments, SNP treatment alone was found more effective in comparison with other treatments under non-stressed and water-stressed conditions. A greater treatment-induced decrease in the H_2_O_2_ level was found in cv. S-24 in comparison with cv. Fsd-2008 (Figure 3G; Table 1).

### 3.4. Plant Water Relations and Yield Attributes of Water-Stressed Wheat Plants Grown from Seeds Primed with Different Levels of SNP and H_2_O_2_

Imposition of water stress also imposed negative impacts on leaf Ψ_s_ and Ψ_w_ of both wheat cultivars. Seed priming with SNP and H_2_O_2_ alone and in combination (SNP + H_2_O_2_) significantly improved the leaf Ψ_s_ and Ψ_w_ under both water-stressed and non-stressed conditions in both wheat cultivars. In relation with leaf Ψ_s_ and Ψ_w_, SNP treatment alone was more useful followed by H_2_O_2_ under both water-stressed and non-stressed conditions (Figure 4A,B; Table 1).

LRWC and Ψ_p_ were also reduced significantly in both studied wheat cultivars, with the lowest decrease found in cv. S-24 in comparison with cv. Fsd-2008 under water stress. However, seed priming with SNP and H_2_O_2_ alone and in combination (SNP + H_2_O_2_) was helpful in improving the LRWC and Ψp of both cultivars of wheat. Among different treatments, H_2_O_2_ treatment alone was more effective under both non-stressed and water-stressed conditions in both studied wheat cultivars. Treatment-induced maximal improvement in LWRC and Ψp was found in cv. S-24 in comparison with cv. Fsd-2008 under water stress (Figure 4C,D; Table 1).

A significant (*p* ≤ 0.001) reduction was found in the number of tillers in plants of both wheat cultivars due to water stress. Seed priming with SNP and H_2_O_2_ alone and in combination (SNP + H_2_O_2_) was effective in improving the number of tillers in both studied wheat cultivars. Among different treatments, SNP treatment was more effective in comparison with other treatments under both water-stressed and non-stressed conditions (Figure 4E; Table 1).

The 100 grain weight and grain yield per plant was also decreased considerably under drought stress, and this decrease was similar in both wheat cultivars. Seed priming with SNP and H_2_O_2_ alone and in combination (SNP + H_2_O_2_) was effective in decreasing the deleterious effects of water stress on the 100 grain weight and grain yield per plant in wheat plants of both cultivars. Seed priming with SNP was more effective followed by H_2_O_2_ and SNP + H_2_O_2_ in enhancing the hundred grain weight and grain yield in plants of both cultivars under both non-stressed and water-stressed conditions (Figure 4F,G; Table 1).

### 3.5. PCA Analysis Extracted from XLSTAT Software of All the Studied Attributes of Wheat Plants Grown Under Water-stressed Conditions from Seeds Primed with Different Levels of SNP and H_2_O_2_

Correlation studies shows that significant positive correlations were found among different growth and yield attributes, T P, LRWC, TPC, TSP, and leaf photosynthetic pigments, but a negative correlation of all these attributes was found with W P, O P, lipid peroxidation, and enzymatic antioxidants (Figure 5; Table 2). Data presented in Figure 5 as well as in Table 2 shows that of the extracted components, F1 and F2 contributed maximally in determining the total variance for the studied attributes in PCA. The maximum contribution is of F1 (78.61%), followed by F2 (10.09%). The total variance of these factors is 88.71%, showing a good contribution.

## 4. Discussion

Different methods are being used for induction of the drought tolerance in different plant species or crop cultivars, including the exogenous use of chemicals through different methods [1,7,20,68]. Drought stress is a major environmental constraint that significantly reduces the productivity of crop plants [69]. Water-stress-induced decrease in plant growth and production is the function of disturbed photosynthetic activity due to altered plant water relation, photosynthetic pigments, oxidative damages to macromolecules, membrane impairment, and in some cases inhibition of the enzyme activity [70]. However, plants have the ability to maintain such perturbations in growth and yield by internal physio-biochemical alterations. However, the stress tolerance in a plant species- or cultivar-specific phenomenon. Some high-yielding plant species or cultivars are sensitive to such abiotic adverse conditions. Among different techniques that are being used for the induction of stress tolerance, seed priming is considered an important one [4]. In seed priming, the seeds are soaked in specific levels of a given chemical for a specific time, which is enough for its absorption to the embryonic regions [21]. These absorbed chemicals are then used at the lateral growth stages for the induction of stress tolerance by modulating different metabolic processes [20]. Similarly, Teng et al. [71] and Wang et al. [72] suggested that early exposure to gaseous chemical priming agents (such as NO, H_2_O_2_, ABA, and SA) was found effective in increasing the ability of plants to tolerance the abiotic stresses.

In the present study, water deficit conditions markedly decreased the growth and yield of both wheat cultivars, which is a common, well-known factor under most abiotic stresses, including water stress, and is directly linked to plant photosynthetic activity, including the content of leaf photosynthetic pigments. Drought is known to reduce the plant growth and yield as a result of reduced photosynthetic rate [26], a function of both stomatal and non-stomatal factors, as well as the light capturing ability by photosynthetic pigments to drive the photosynthetic process efficiently. The disturbance in this process also disturbs the assimilation mechanism that is directly involved in better seed production [73]. Our findings are similar to [74,75], who reported a drought-triggered reduction in biomass production and yield of maize and canola, respectively. However, seed priming with signaling molecules improved the biomass and yield of both studied wheat cultivars under water-stressed conditions. Ashraf [13] reported that H_2_O_2_ seed pre-treatment increased the growth of drought-stressed maize plants. They found that this beneficial role of H_2_O_2_ seed priming on growth under water-deficit conditions was due to its signaling properties [76]. Furthermore, while working with water-stressed canola plants, [75] reported that low doses of H_2_O_2_ acted as signaling molecules that increased the growth and yield, which was associated with their improved antioxidative defense mechanism. Increase in the levels of endogenous nitric oxide due to exogenously-applied SNP has been revealed to increase the net photosynthetic rate in plants exposed to osmotic stress [77], and they concluded that this increase was related to an increase in leaf chlorophyll contents. Higher photosynthetic rates were related positively with increased growth of plants [74,78]. Ashraf [13] reported a similar association among the photosynthesis rate and biomass production in water-stressed maize plants when H_2_O_2_ was applied as a seed treatment.

In the present investigation, a significant decrease was recorded in leaf chlorophyll contents of both wheat cultivars when grown under water stress. Similar to our findings, the decrease in chlorophyll biosynthesis due to drought stress has been observed among several crops, such as wheat [79], soybean [29], maize [13], and cucumber [26]. However, application of exogenous SNP, H_2_O_2_, and SNP + H_2_O_2_ improved the leaf chlorophyll content of both studied wheat cultivars under both control and water-stressed conditions. NO-induced improvement in leaf chlorophyll biosynthesis might have been due to its potential to increase the leaf chlorophyll content and to protect the chloroplastic membrane containing chlorophyll. Similarly, the role of H_2_O_2_ in enhancing the photosynthetic pigments is due to its ability to stimulate the antioxidative defense system [76,80] as acting signaling molecules. NO- and H_2_O_2_-induced improvement in photosynthetic pigments has also been observed in drought-stressed marigold plants [45]. Similar results were found in the present study, where seed priming with NO and H_2_O_2_ improved the photosynthetic pigments of wheat plants, which might have improved the leaf photosynthetic activity due to greater absorption of light as a result of better biosynthesis of photo-assimilates that improved the growth and yield of water-stressed wheat plants.

ROS-induced oxidative damage is a common phenomenon under drought stress that leads to membrane lipids peroxidation at cellular and subcellular levels and decreases the growth and biomass production along with seed yield. Lipid peroxidation is measured in terms of MDA accumulation, as well as some other biochemical attributes [81]. In the present investigation, increased levels of MDA and H_2_O_2_ were found in wheat plant cultivars under water stress. Similar to the present study, Ali et al. [81] and Nahar et al. [82] also found an increased accumulation in MDA and H_2_O_2_ in mung bean plants under drought stress. However, in the present study, seed priming treatment with SNP, H_2_O_2_, and SNP + H_2_O_2_ effectively reduced the MDA and H_2_O_2_ accumulation in both wheat cultivars grown under drought stress. It was found by Zhang et al. [43] that seed priming with NO can decrease the levels of H_2_O_2_ and lipid peroxidation when grown under water stress. Furthermore, it has been proposed that under water-stressed conditions, NO itself reacts with ROS as a chain terminator to suppress the peroxidation of membrane lipids [83]. Ashraf [16], while working with drought-treated maize plants, reported that seed priming with H_2_O_2_ reduced the levels of MDA and also of H_2_O_2_ by improving the antioxidative defense mechanism, which neutralized the over-produced ROS and protected the membrane against oxidative impairments. This reduced lipid peroxidation can be attributed to better growth and yield, because chlorophyll as an integral part of chloroplastic membranes increased significantly in the present study, which correlates well with the growth and seed yield. This might be due to reduced lipid peroxidation of chloroplastic membranes as a result of better functioning of chloroplastic membranes, leading to better photosynthesis. However, no information is available in literature demonstrating the role of combined application of SNP and H_2_O_2_ in reducing the MDA and H_2_O_2_ accumulation.

The antioxidative defense system of plants helps in reducing the stress-induced ROS accumulation, which comprises the enzymatic (CAT, POD, and SOD) and non-enzymatic (ascorbic acid, phenolics, flavonoids, and proline) scavengers [81,84,85]. The results of the present experiment showed that the activities of enzymatic antioxidants, such as CAT, POD, and SOD, were enhanced in both wheat cultivars in response to water stress. Drought induced this improvement in the activities of enzymatic antioxidants, such as POD, CAT, and SOD, which has previously been reported by Wang et al. [23] in grapes plant under stress conditions. In the present experiment, a further improvement in the activities of enzymatic antioxidants, such as CAT, SOD, and POD, was found in both studied cultivars of wheat due to seed priming with SNP and H_2_O_2_ alone and in combination under control and drought-stressed conditions, signifying that exogenous application of SNP and H_2_O_2_ helped in regulating the over-produced ROS to maintain the membrane’s integrity, mostly by improving the activities of key enzymatic antioxidants such as CAT, SOD, and POD under water-stressed conditions. Moussa and Mohamed [56] reported that seed priming with H_2_O_2_ or SNP improved the activities of enzymatic antioxidants in drought-stressed pea plants. It has been found that H_2_O_2_ pre-treatment stimulates the antioxidative defense mechanism in several plants exposed to multiple stresses, such as drought [13], salinity [25], and heavy metal stress [86]. However, reports showing the combined influences of H_2_O_2_ and NO in improving the antioxidative defense mechanism are not available.

Furthermore, cellular metabolites such as leaf phenolics and ascorbic acid contents help in regulating several plants metabolic processes [8], including playing a role in reducing ROS-induced oxidative impairment. In the present experiment, exogenous treatment of SNP, H_2_O_2_, and SNP + H_2_O_2_ was helpful in improving the leaf phenolic and ascorbic acid contents in water-stressed wheat plants. SNP-triggered increased production of phenolic and AsA contents might be due to the effective role of NO in the stimulation of the antioxidant defense system [52]. Habib et al. [8] reported that seed priming with SNP was effective in improving the leaf phenolic and AsA contents in salt-stressed rice plants. Similarly, H_2_O_2_ pre-treatment also improved the leaf AsA content considerably in wheat plants grown under water-deficit conditions [87].

In order to deal with the deleterious impacts of water stress, plants undergo osmotic regulation by increasing the synthesis of potential osmolytes (GB, proline, and total soluble proteins) in the cytosol and in other organelles. GB and proline are the key osmolytes that contribute significantly in cellular osmotic adjustment. Leaf proline, an important secondary metabolite, performs dual functions in plants as an antioxidant as well as an osmoprotectant [8]. As an antioxidative compound, it has the potential to protect the membranes of the organelles, and as a osmolyte it plays a key role in improving cellular water relations under water-stressed conditions [20,88]. It is well known that under osmotic stress, a high concentration of proline acts as a water substitute to stabilize the cellular structures through their hydrophobic interactions and hydrogen bonding, which protects the membranes from dehydration [89]. GB has the potential to activate stress-related genes, detoxify ROS, and protect photosynthetic machinery, as well as maintain structures of proteins by acting as molecular chaperones under stressed conditions [90]. In the current experiment, a similar increasing trend in cellular GB, proline, and total soluble protein contents was observed in both studied wheat cultivars as a result of exogenously applied SNP, H_2_O_2_, and SNP + H_2_O_2_ as seed primers, which induced a greater protection to plants under stressed conditions. NO-induced enhanced accumulation of these osmolytes correlates well with the findings for several plant species that have already been studied [44,91,92]. As a ROS scavenger, proline acts as a non-enzymatic antioxidant of ^1^ O_2_ and OH. Alia et al. [93] demonstrated that proline acts as a powerful ^1^ O_2_ quencher, or converts it to other types of ROS, such as superoxide radicles and peroxide anions. The superoxide radical is then scavenged by the SOD to generate the H_2_O_2_. Furthermore, the proline metabolism produces other stress-related compounds that are directly involved in the antioxidative defense mechanism [33]. Therefore, in the present study, increased accumulation of proline might have acted as a direct scavenger of ROS or boosted the antioxidation mechanism by playing a key role as a signaling molecule [33]. As a result, it was effective in maintaining better stabilization of sub-cellular structures and membranes, stabilization of proteins, as well as the maintenance of cellular functions.

NO^−^ and H_2_O_2_ triggered increased accumulation of proline, GB, and total soluble proteins, which may improve the drought tolerance ability of wheat plants through osmotic adjustment by maintaining better cellular water content, as a result leading to better growth of wheat plants. Liao et al. [30] reported that exogenous application of NO or H_2_O_2_ indirectly can improve the TSP content in marigold plants grown under water stress. As a result, reduced osmotic potential was associated with improved water uptake, and consequently relative water content, plant biomass production, and seed yield. Moreover, in the present investigation, the mitigation of the deleterious effects of water stress on plant growth and yield after pretreatment with SNP could be related to the enhancement in proline accumulation. Our results are in agreement with works reporting concomitant NO and proline production in cucumber and rice under drought stress [34,94]. In *Cakile maritima* seedlings grown under water stress after pretreatment with SNP, the donor of NO significantly improved the proline accumulation, and this improvement in proline accumulation was associated with the increased activity of P5CS, a proline synthesizing enzyme [48]. Furthermore, Shi et al. [95] reported that the transgenic lines over-expressing NOS in *Arabidopsis thaliana* accumulated more NO and produced more proline as compared with wild type. It has also been found that in corn seedlings grown under osmotic stress, greater accumulation of proline in leaves was due to H_2_O_2_ treatment, which was found due to an almost immediate and rapid increase in the activities of arginase and ornithine aminotransferase (OAT), glutamate dehydrogenase (GDH), and ∆1-pyrroline-5-carboxylatesynthetase (P5CS), with an alternate decrease in the activity of proline dehydrogenase [27]. However, reports revealing the combined influence of NO and H_2_O_2_ in increasing the accumulation of osmolytes are not available.

In the present study, water relation attributes, such as leaf Ψs, Ψw, Ψp, and LRWC, were negatively affected by water stress in both studied wheat cultivars. Similar results were reported by Abid et al. [70] in drought-treated plants. However, plants grown from SNP, H_2_O_2_, and SNP+H_2_O_2_ pre-treated seeds showed significant less reduction in leaf water relation parameters under water-deficit conditions. Habib and Ashraf [37] reported that in drought-stressed rice plants, NO pre-treatment improved the Ψs and Ψw, and as a result improved Ψp and LRWC. Improved LRWC due to H_2_O_2_ pre-treatment was observed in soyabean plants under drought stress [22]. This improvement in cellular water relations due to NO and H_2_O_2_ treatment might be a function of greater accumulation of compatible solutes, such as proline and GB, which effectively lowered the cellular osmotic potential as a result of greater uptake of water by roots from soil, and maintained better Ψw and Ψp. The latter factors are necessary for better growth due to the involvement of plant water relations in regulating the photosynthetic stomatal factors for better water use efficiency with improved photosynthesis, which improves the growth and final yield.

From the findings it can be concluded that seed priming with H_2_O_2_ and SNP alone or in combination significantly improved the growth and grain yield of water-stressed wheat plants, which was associated with increased biosynthesis of photosynthetic pigments, as well as the improvement in leaf water relations through cellular osmotic adjustment of the accumulation of osmolytes. The induction of water stress tolerance in wheat plants after pretreatment with NO and H_2_O_2_ was also associated with the antioxidative defense mechanism through the improvement in the activities of antioxidant enzymes and accumulation of non-enzymatic antioxidants, which effectively reduced the membrane lipid peroxidation. As a result, better functioning of the cellular membranes occurred, which led to better growth and yield of water-stressed wheat plants.

## Figures and Tables

**Figure 1 plants-09-00285-f001:**
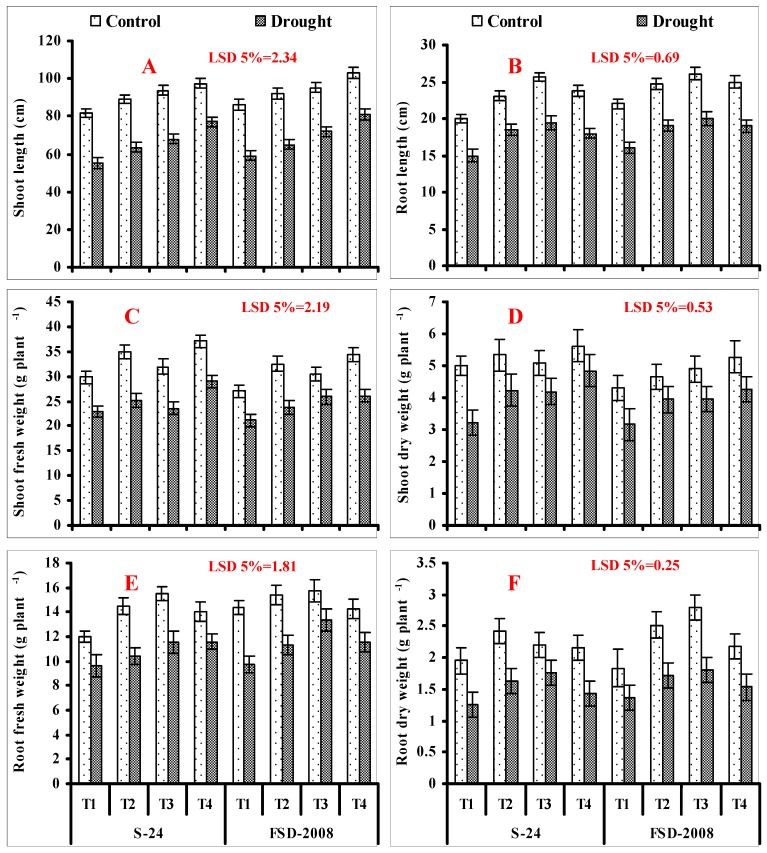
Shoot length (**A**), root length (**B**), shoot fresh weight (**C**), shoot dry weight (**D**), root fresh weight (**E**), and root dry weight (**F**) of wheat plants grown from non-primed (T1) seeds and seeds primed with sodium nitroprusside (SNP) (T2), H_2_O_2_ (T3), and SNP + H_2_O_2_ (T4) under non-stressed and water-stressed conditions {(mean ± standard error (SE)}.

**Figure 2 plants-09-00285-f002:**
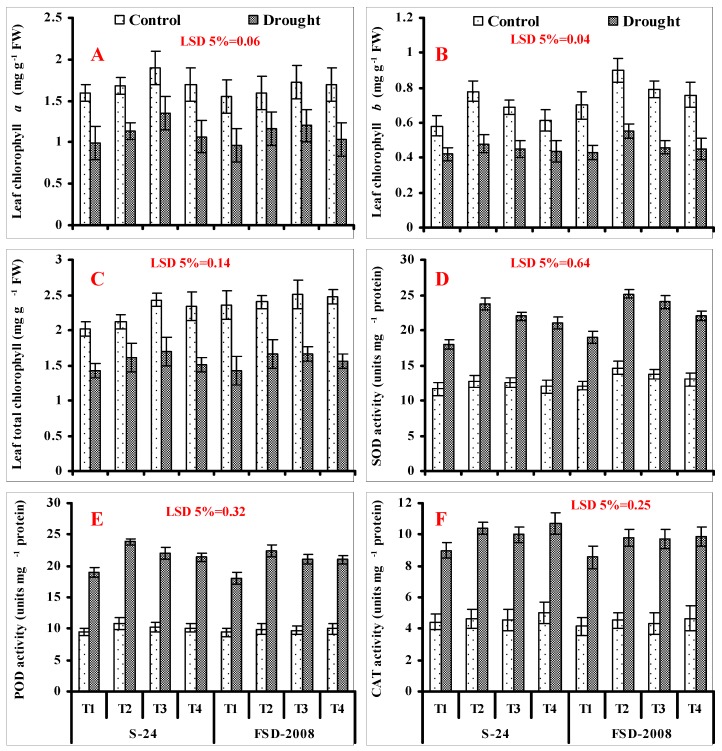
Leaf chlorophyll *a* (**A**), chlorophyll *b* (**B**), total chlorophyll (**C**), Superoxide dismutase activity (**D**), Peroxidase activity (**E**), and Catalase activity (**F**) of wheat plants grown from non-primed seeds (T1) and seeds primed with SNP (T2), H_2_O_2_ (T3), and SNP + H_2_O_2_ (T4) under non-stressed and water-stressed conditions (mean ± SE).

**Figure 3 plants-09-00285-f003:**
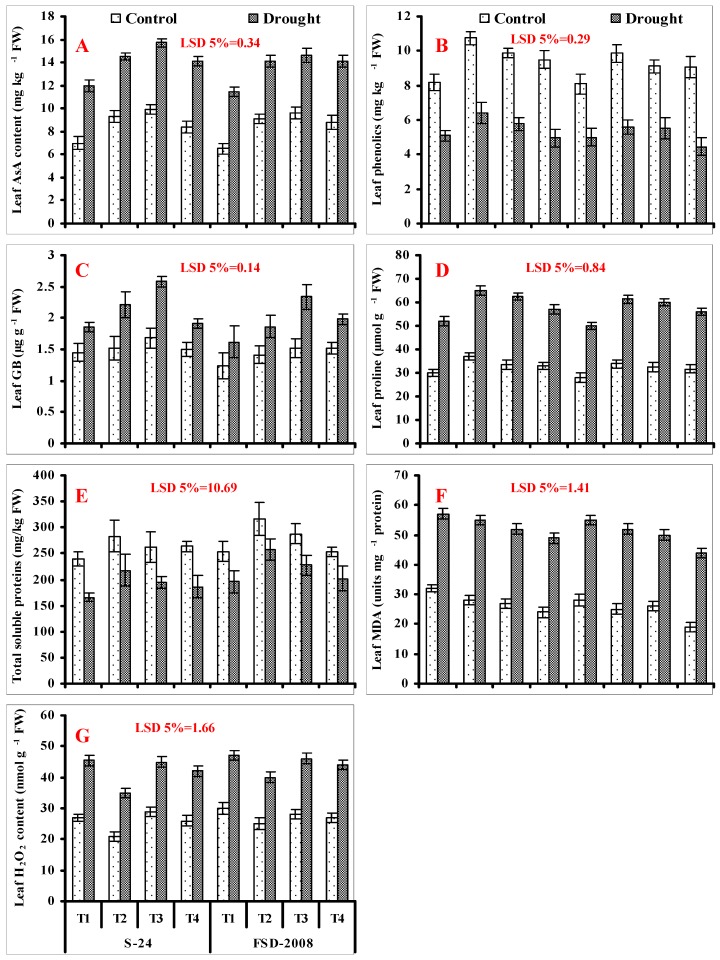
Leaf AsA (**A**), phenolics (**B**), glycine betaine (GB) (**C**), proline (Pro) (**D**), total soluble protein (TSP) (**E**), leaf malondialdehyde (MDA) (**F**), and leaf H_2_O_2_ (**G**) contents of wheat plants grown from non-primed seeds (T1) and seeds primed with SNP (T2), H_2_O_2_ (T3), and SNP + H_2_O_2_ (T4) under non-stressed and water-stressed and conditions (mean ± SE).

**Figure 4 plants-09-00285-f004:**
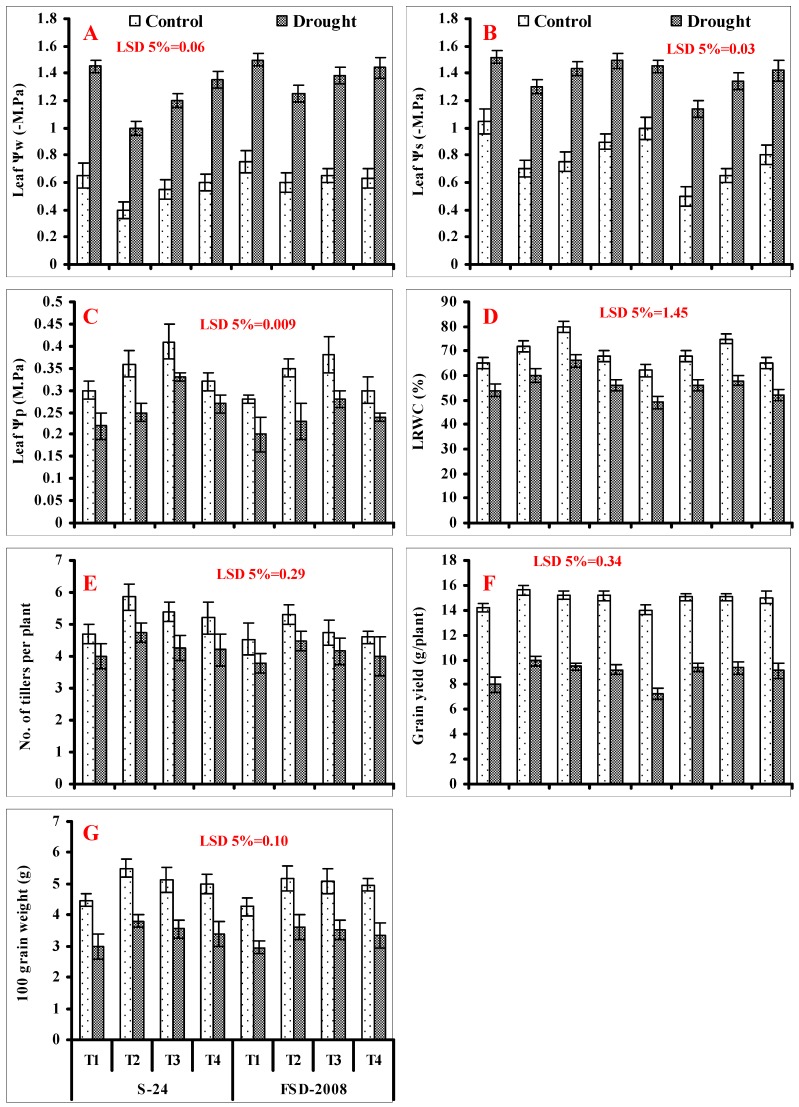
(**A**) Leaf water potential (Ψ_w_), (**B**) leaf turgor potential (Ψ_p_), (**C**) osmotic potential (Ψ_s_), (**D**) leaf relative water content (LRWC), number of tillers (**E**), grain yield per plant (**F**), and 100 grain weight (**G**) of wheat plants grown from non-primed seeds (T1) and seeds primed with SNP (T2), H_2_O_2_ (T3), and SNP + H_2_O_2_ (T4) under non-stressed and water-stressed conditions (mean ± SE).

**Figure 5 plants-09-00285-f005:**
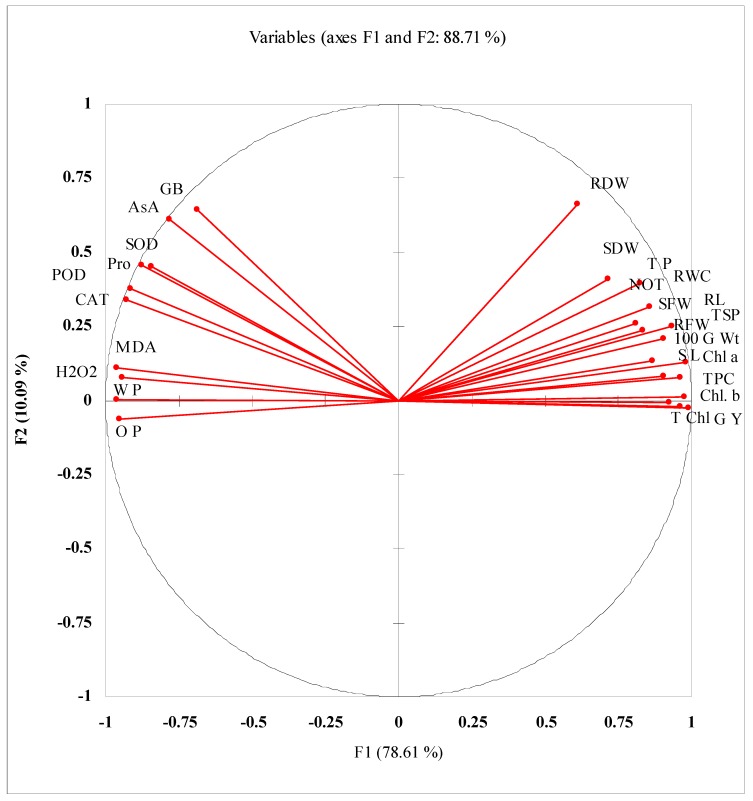
Principal component analysis (PCA) of different studied parameters of wheat plants grown from seeds after priming with SNP, H_2_O_2_, and SNP + H_2_O_2_ under non-stressed and water-stressed conditions.

**Table 1 plants-09-00285-t001:** Mean squares from analysis of variance of the data for different studied attributes.

**SOV**	**df**	**SFW**	**SDW**	**RFW**	**RDW**	**SL**	**RL**	**Yield/Plant**
Stress (S)	1	183.39 ***	2.830 **	43.99 **	0.507 **	2440 ***	129.62 ***	141.50 ***
Cultivars CV)	1	11.27 *	0.628 *	2.584 *	0.131 *	45.76 **	4.45 **	0.39 *
Treatments (T)	3	44.65 **	1.312 *	4.464 *	0.600 **	258.98 ***	14.74 ***	2.13 **
S × CV	1	1.54 ns	0.047 ns	0.066 ns	6.20 ns	0.01 ns	0.275 ns	0.005ns
S × T	3	5.94 ns	0.012 ns	0.400 ns	0.045 ns	6.510 ns	0.201 ns	0.130 ns
CV × T	3	2.32 ns	0.062 ns	0.245 ns	0.039 ns	1.462 ns	0.162 ns	0.042 ns
Error	22	0.94	0.055	0.648	0.012	1.077	0.093	0.023
**SOV**	**df**	**100 Grain wt.**	**No. of Tillers**	**LRWC**	**Leaf Ψs**	**Leaf Ψw**	**Leaf Ψp**	**Chl. *a***
Stress (S)	1	9.672 ***	2.788 ***	676 ***	1.404 ***	2.059 ***	0.0289 ***	1.274 ***
Cultivars CV)	1	0.050 *	0.476*	81 ***	0.044 ***	0.062 **	0.0025 **	0.014 *
Treatments (T)	3	0.526 ***	1.5113 **	112 ***	0.088 ***	0.054 **	0.0069 ***	0.050 **
S × CV	1	0.004 ns	0.090 ns	2.25 ns	2.250^e-4^ ns	0.0012 ns	1.00^e-4^ ns	9.22^e-4^ ns
S × T	3	0.011 ns	0.009 ns	2.83 ns	0.009 **	0.005 ns	6.17^e-4^ **	0.004 ns
CV × T	3	0.010 ns	0.010 ns	1.83 ns	0.003 *	0.006 ns	1.17^e-4^ ns	0.005 ns
Error	22	0.002	0.016 ns	0.42	1.75^e-4^	8.417^e-4^	1.67^e-5^	7.31^e-4^ ns
**SOV**	**df**	**Chl. *b***	**Total Chl.**	**Leaf AsA**	**Phenolics**	**GB**	**Proline**	**TSP**
Stress (S)	1	0.285 ***	2.332 ***	110.14 ***	62.52 ***	1.276 ***	2608 ***	16320 ***
Cultivars CV)	1	0.021 **	0.049 *	0.448 *	0.916 **	0.093 *	17.49 **	2047 **
Treatments (T)	3	0.015 **	0.049 *	7.713 ***	1.860 **	0.162 **	64.58 ***	2249 **
S × CV	1	0.009 *	0.038 *	0.144ns	0.018 ns	0.006 ns	0.16 ns	203.0 6ns
S × T	3	0.002 ns	0.011 ns	0.071ns	0.426 *	0.044 *	8.93 **	3.56 ns
CV × T	3	3.10^e-4^ ns	0.005 ns	0.16ns	0.092 ns	0.017ns	0.87 ns	215.06 *
Error	22	3.18^e-4^	0.004	0.023	0.016	0.003	0.14	22.56
**SOV**	**df**	**POD**	**CAT**	**SOD**	**MDA**	**H_2_O_2_**		
Stress (S)	1	493.83 ***	109.15 ***	29.42 ***	2626 ***	1080 ***		
Cultivars CV)	1	1.7755 **	0.60 **	6.32 **	39.06 **	17.28 *		
Treatments (T)	3	5.3091 ***	0.77 **	10.99 **	6.062 ***	42.85 **		
S × CV	1	0.262 *	0.08 ns	0.04ns	0.062 ns	0.432 ns		
S × T	3	2.378 **	0.24 *	3.155 **	1.229 ns	1.913 ns		
CV × T	3	0.130 ns	0.02 ns	0.233 ns	2.062 ns	3.479 ns		
Error	22	0.020	0.012	0.08055	0.396	0.546		

Note: *, ** and *** = significant at 0.05, 0.01 and 0.001 levels respectively. SFW = shoot fresh weight; SDW = shoot dry weight; RFW = root fresh weight; RDW = root dry weight; SL = shoot length; RL = root length; GY = grain yield per plant; 100 GW = 100 grain weight; NOT = number of tillers per plant; RWC = relative water content; O P = osmotic potential; W P = water potential; T P = turgor potential; Chl. *a* = leaf chlorophyll *a*; Chl. *b* = leaf chlorophyll *b*; T Chl.= leaf total chlorophyll; AsA = ascorbic acid; TPC = total phenolic content; GB = glycine betain; TSP = total soluble proteins; POD = peroxidase; CAT = catalase; SOD = superoxide dismutase; MDA = malondialdehyde.

**Table 2 plants-09-00285-t002:** Spearman correlation coefficients (*r*^2^) values of growth, yield, and water relation attributes with the all studied attributes.

Variables	SFW	SDW	RFW	RDW	SL	RL	GY	100 GW	NOT
SFW	1	0.910 ***	0.735 ***	0.629 ***	0.861 ***	0.786 ***	0.798 ***	0.842 ***	0.767 ***
SDW	0.910 ***	1	0.658 ***	0.593 ***	0.815 ***	0.745 ***	0.719 ***	0.761 ***	0.695 ***
RFW	0.735 ***	0.658 ***	1	0.760 ***	0.885 ***	0.959 ***	0.879 ***	0.878 ***	0.708 ***
RDW	0.629 ***	0.593 ***	0.760 ***	1	0.597 ***	0.773 ***	0.555 ***	0.673 ***	0.574 ***
SL	0.861 ***	0.815 ***	0.885 ***	0.597 ***	1	0.923 ***	0.900 ***	0.872 ***	0.646 ***
RL	0.786 ***	0.745 ***	0.959***	0.773 ***	0.923 ***	1	0.912 ***	0.925 ***	0.740 ***
GY	0.798 ***	0.719 ***	0.879 ***	0.555 ***	0.900 ***	0.912 ***	1	0.976 ***	0.837 ***
100 GW	0.842 ***	0.761 ***	0.878 ***	0.673 ***	0.872 ***	0.925 ***	0.976 ***	1	0.903 ***
NOT	0.767 ***	0.695 ***	0.708 ***	0.574 ***	0.646 ***	0.740 ***	0.837 ***	0.903 ***	1
RWC	0.678 ***	0.697 ***	0.822 ***	0.672 ***	0.718 ***	0.862 ***	0.852 ***	0.882 ***	0.822 ***
O P	−0.737 ***	−0.595 ***	−0.884 ***	−0.689 ***	−0.817 ***	−0.917 ***	−0.928 ***	−0.956 ***	−0.833 ***
W P	−0.757 ***	−0.698 ***	−0.800 ***	−0.520 ***	−0.812 ***	−0.856 ***	−0.977 ***	−0.972 ***	−0.892 ***
T P	0.690 ***	0.693 ***	0.854 ***	0.739 ***	0.737 ***	0.859 ***	0.806 ***	0.843 ***	0.770 ***
Chl a	0.742 ***	0.715 ***	0.881 ***	0.592 ***	0.861 ***	0.925 ***	0.962 ***	0.944 ***	0.803 ***
Chl b	0.679 ***	0.517 ***	0.869 ***	0.641 ***	0.796 ***	0.879 ***	0.895 ***	0.911 ***	0.763 ***
T Chl	0.718 ***	0.641 ***	0.916 ***	0.589 ***	0.903 ***	0.952 ***	0.953 ***	0.917 ***	0.706 ***
AsA	−0.479 ***	−0.312 *	−0.582 ***	−0.079 ns	−0.653 ***	−0.574 ***	−0.797 ***	−0.692 ***	−0.514 ***
TPC	0.790 ***	0.677 ***	0.837 ***	0.589 ***	0.809 ***	0.874 ***	0.967 ***	0.981 ***	0.905 ***
GB	−0.445 **	−0.248 ns	−0.500 ***	−0.040 ns	−0.583 ***	−0.486 ***	−0.674 ***	−0.595 ***	−0.440 **
Proline	−0.596 ***	−0.443 **	−0.724 ***	−0.248 ns	−0.786 ***	−0.720 ***	−0.872 ***	−0.783 ***	−0.566 ***
TSP	0.645 ***	0.512 ***	0.846 ***	0.724 ***	0.729 ***	0.860 ***	0.841 ***	0.879 ***	0.793 ***
POD	−0.640 ***	−0.482 ***	−0.768 ***	−0.339 *	−0.803 ***	−0.767 ***	−0.906 ***	−0.836 ***	−0.632 ***
CAT	−0.618 ***	−0.473 ***	−0.783 ***	−0.371 **	−0.791 ***	−0.791 ***	−0.925 ***	−0.864 ***	−0.676 ***
SOD	−0.613 ***	−0.478 ***	−0.650 ***	−0.184 ns	−0.734 ***	−0.655 ***	−0.850 ***	−0.767 ***	−0.597 ***
MDA	−0.802 ***	−0.702 ***	−0.879 ***	−0.515 ***	−0.961 ***	−0.909 ***	−0.961 ***	−0.916 ***	−0.698 ***
H_2_O_2_	−0.772 ***	−0.682 ***	−0.756 ***	−0.469 ***	−0.805 ***	−0.810 ***	−0.961 ***	−0.954 ***	−0.879 ***

Note: *, ** and *** = significant at 0.05, 0.01 and 0.001 levels respectively.

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
