# Peer review of "Use of Nitric Oxide and Hydrogen Peroxide for Better Yield of Wheat (Triticum aestivum L.) under Water Deficit Conditions: Growth, Osmoregulation, and Antioxidative Defense Mechanism"

_plants, 2020, doi:10.3390/plants9020285_

Round 1

Reviewer 1 Report

In this manuscript, authors studied the usefulness of nitric oxide and hydrogen peroxide as priming agents to mitigate drought stress in wheat.  This is an interesting study and deserves to be published in plants. However, this manuscript requires major revision before publication. English needs professional editing. Methods description is too shallow. For example, no description of how long the plants were stressed and how 60% field capacity was maintained. No description of when the assays were done. What was the growth stage of plants during stress imposition? Figures: What is T1, T2, T3 and T4? For better understanding, it is nice to put same genotype and different treatments together rather than different treatments together. Effect of drought stress on plant growth and development needs to be reduced in all the sections as this is already known. Description should be more on the role of nitric oxide and hydrogen peroxide as priming agents.

Author Response

Reviewer 1

Comment: This is an interesting study and deserves to be published in plants

Response: Dear reviewer thanks for good comments regarding the manuscript publication

Comment: English needs professional editing.

Response: English language of the manuscript has been improved significantly

Comment: Methods description is too shallow

Response: Now the methodology has been improved and given in detailed

Comment: For example, no description of how long the plants were stressed and how 60% field capacity was maintained

Response: Now detailed information has been included in the manuscript highlighted in red 

Comment: No description of when the assays were done

Response: Information regarding the assays has been included

Comment: What was the growth stage of plants during stress imposition?

Response: Information added in text in Materials and methods section

Comment: Figures: What is T1, T2, T3 and T4?

Response: Detailed information regarding the given treatments has been incorporated below the each Figure highlighted in red

Comment: For better understanding, it is nice to put same genotype and different treatments together rather than different treatments together.

Response: Suggestion has been followed. Now the pattern of the Figures has been revised following the suggestions of the worthy Reviewer 

Comment: Effect of drought stress on plant growth and development needs to be reduced in all the sections as this is already known. Description should be more on the role of nitric oxide and hydrogen peroxide as priming agents.

Response: Suggestion has been followed

Reviewer 2 Report

The manuscript entitled: “Use of nitric oxide and hydrogen peroxide for better yield of wheat (Triticum aestivum L.) under water deficit conditions: Growth, osmoregulation and antioxidative defence mechanism” describes different treatments of seed soaking under water stress conditions in order to improve tolerance of wheat.

I have some minor modifications which are presented below:

Line 71: Rephrase as “In order to reduce

Line 131: Why do you study these two cultivars? Did you have any indication that are less drought tolerant?

Line 198-199: What do you mean by “while in case of shoot length the combined treatment (SNP+H2O2) was found more helpful under drought stress?” Please rewrite the sentence

Line 269: please put a separator for the two words: theleaf

Author Response

Reviewer 2

Comment: Line 71: Rephrase as “In order to reduce

Response: Corrected

Comment: Line 131: Why do you study these two cultivars? Did you have any indication that are less drought tolerant?

Response: Required information regarding the use of these two wheat cultivars has been incorporated

Comment: Line 198-199: What do you mean by “while in case of shoot length the combined treatment (SNP+H2O2) was found more helpful under drought stress?” Please rewrite the sentence

Response: Suggestion followed

Comment: Line 269: please put a separator for the two words: theleaf

Response: Corrected

Reviewer 3 Report

The submitted manuscript, “Use of nitric oxide and hydrogen peroxide for better yield of wheat (Tritricum aestivum L.) under water deficit conditions: Growth, osmoregulation and antioxidative defense mechanism” reports data about the effect of priming of wheat seeds with NO and H2O2 on water stressed plants.

I have the following recommendations:

- All presented figures are not accompanied with figure legends with a detailed description of the abbreviations used. It makes very difficult to follow the results. I guess that T1 states for control, not primed seeds, T2 – for primed with NO, T3 – primed with H2O2 and T4 – for primed with both substances but it has to be described in the figure caption.

- Line 155 – in the formula for calculation of LRWC – in the denominator “leaf” is missing before “dry weight”.

- The authors state that they have a statistical analysis, but in the text is discussed only the statistical difference between watered and water stressed plants.

- My main concern is that priming of seeds with NO, H2O2 or with combination of both agents leads to similar effects in normally watered and water-stressed plants. Thus it is difficult to state that priming is affecting positively the water-stressed plants. 

- Language needs significant improvement.

My opinion is that the manuscript needs major revision before accepting for publishing.

Author Response

Reviewer 3

Comment: All presented figures are not accompanied with figure legends with a detailed description of the abbreviations used. It makes very difficult to follow the results. I guess that T1 states for control, not primed seeds, T2 – for primed with NO, T3 – primed with H2O2 and T4 – for primed with both substances but it has to be described in the figure caption.

Response: Suggestion has been followed and now the information regarding the treatments has been incorporated in detailed below each Figure

Comment: - Line 155 – in the formula for calculation of LRWC – in the denominator “leaf” is missing before “dry weight”.

Response:

Comment: - The authors state that they have a statistical analysis, but in the text is discussed only the statistical difference between watered and water stressed plants.

Response: Now the detail regarding the statistical analysis among treatments has been incorporated in the text and a detailed table has been impeded showing the significant differences among different treatments as well as the cultivars

Comment: - My main concern is that priming of seeds with NO, H2O2 or with combination of both agents leads to similar effects in normally watered and water-stressed plants. Thus it is difficult to state that priming is affecting positively the water-stressed plants. 

Response: Dear reviewer though the improvements have also been recorded in non-stressed plants but not in all studied attributes. Furthermore, this improvement in the studied attributes is comparatively less in non-stressed conditions as compared with water stressed plants.

Comment: - Language needs significant improvement.

Response: Now the language has been improved following the suggestions

Round 2

Reviewer 3 Report

The revised manuscript “Use of nitric oxide and hydrogen peroxide for better yield of wheat (Triticum aestivum L.) under water deficit conditions: Growth, osmoregulation and antioxidative defence mechanism” is in much better shape than the submitted manuscript.

There are still things that have to be added before accepting the manuscript for publication.

All figures need the correct description and numbering. Above the figure should be the number of the figure – Figure 1, 2, … etc, and below every figure the caption has be self-descriptive – every panel has to contain A, B, C… and in the caption to be described on every panel what parameter is includes. Description of T1, T2, T3, T4 is not enough. Table 1 and Table 2 – List of abbreviation is given below Table 2. It should be given after Table 1 in order the reader to be aware what is following. For me it is strange to give the results of statistic analysis of all values separately. It makes the evaluation of statically significant differences between presented data very difficult. The language is corrected in major part of manuscript but sill there several points left: Line 47 – “…to…” to be substituted by “….by…..” Line 52 – “… damages….” To be substituted by “…. Damage…” Line 61 – “… stressed…” to be substituted “…stressful…” Line 65 – unclear sentence, needs correction. Line 68 – “in ordor..” to be substituted by “ In order”. Line 82 – “…H2O2..” 2 to be substript. Line 111 – before “both” to be removed “the”. Line 246 – “Bomin” to be substituted by “Bovin”. In the section 2.2 Activities of enzymatic antioxidants – to check and correct the indications of all concentrations - for example - Line 201 – “mm phosphate buffer”. Line 249 – “… enzyme…” probably the authors meen extract. The description in the text – Line 338-345 does not correspond to the results presented in the panel with data about the amount of GB. Line 347-350 – the sentence needs revision.

Author Response

Comment: All figures need the correct description and numbering. Above the figure should be the number of the figure – Figure 1, 2, … etc, and below every figure the caption has be self-descriptive – every panel has to contain A, B, C… and in the caption to be described on every panel what parameter is includes. Description of T1, T2, T3, T4 is not enough.

Response: Suggestions have been taken. The figures are now labeled and each panel in figure against each parameter has been labeled with a specific alphabet and the self descriptive detail of each parameter with the specific alphabet has been provided below each figure. All changes are highlighted in red. The changes have also been made in the text against each parameter regarding the numbering and labeling. 

Comment: Table 1 and Table 2 – List of abbreviation is given below Table 2. It should be given after Table 1 in order the reader to be aware what is following.

Response: Suggestion has been taken

Comment: For me it is strange to give the results of statistic analysis of all values separately. It makes the evaluation of statically significant differences between presented data very difficult.

Response: For the better understanding of significant differences LSD values for each parameter regarding each parameter has been given in each panel highlighted in red

Comment: The language is corrected in major part of manuscript but sill there several points left:

Response: Thankful for the worthy reviewer for appreciating in improving the English language and further has been improved highlighted in red

Comment: Line 47 – “…to…” to be substituted by “….by…..”

Response: Corrected

Comment: Line 52 – “… damages….” To be substituted by “…. Damage…”

Response: Corrected

Comment: Line 61 – “… stressed…” to be substituted “…stressful…”

Response: Corrected

Comment: Line 65 – unclear sentence, needs correction.

Response: Correction has been made. I hope now the sentence will be clear

Comment: Line 68 – “in ordor..” to be substituted by “ In order

Response: Corrected

Comment: Line 82 – “…H2O2..” 2 to be in subscript.

Response: Corrected

Comment: Line 111 – before “both” to be removed “the”.

Response: Corrected

Comment: Line 246 – “Bomin” to be substituted by “Bovin”

Response: Corrected

Comment: In the section 2.2 Activities of enzymatic antioxidants – to check and correct the indications of all concentrations -

Response: Checked and corrected, highlighted in red

Comment: Line 201 – “mm phosphate buffer”. Line 249 – “… enzyme…” probably the authors meen extract

Response: Corrected

Comment: Line 249 – “… enzyme…” probably the authors mean extract

Response: Corected

Comment: The description in the text – Line 338-345 does not correspond to the results presented in the panel with data about the amount of GB.

Response: The part regarding the leaf GB content has been rewritten following the results presented in the panel

Comment: Line 347-350 – the sentence needs revision.

Response: Now the sentence has been revised carefully